# Infinite Time Horizon Safety of Bayesian Neural Networks

**Mathias Lechner**[*]
IST Austria
Klosterneuburg, Austria
mlechner@ist.ac.at

**Đorđe Žikelić**[*]
IST Austria
Klosterneuburg, Austria
dzikelic@ist.ac.at

**Krishnendu Chatterjee**
IST Austria
Klosterneuburg, Austria
kchatterjee@ist.ac.at

**Thomas A. Henzinger**
IST Austria
Klosterneuburg, Austria
tah@ist.ac.at

## Abstract

Bayesian neural networks (BNNs) place distributions over the weights of a neural network to model uncertainty in the data and the network's prediction. We consider the problem of verifying safety when running a Bayesian neural network policy in a feedback loop with infinite time horizon systems. Compared to the existing sampling-based approaches, which are inapplicable to the infinite time horizon setting, we train a separate deterministic neural network that serves as an infinite time horizon safety certificate. In particular, we show that the certificate network guarantees the safety of the system over a subset of the BNN weight posterior's support. Our method first computes a safe weight set and then alters the BNN's weight posterior to reject samples outside this set. Moreover, we show how to extend our approach to a safe-exploration reinforcement learning setting, in order to avoid unsafe trajectories during the training of the policy. We evaluate our approach on a series of reinforcement learning benchmarks, including non-Lyapunovian safety specifications.

## 1 Introduction

The success of deep neural networks (DNNs) across different domains has created the desire to apply them in safety-critical applications such as autonomous vehicles [28, 30] and healthcare systems [41]. The fundamental challenge for the deployment of DNNs in these domains is certifying their safety [3]. Thus, formal safety verification of DNNs in isolation and closed control loops [26, 23, 17, 43, 13] has become an active research topic.

Bayesian neural networks (BNNs) are a family of neural networks that place distributions over their weights [36]. This allows learning uncertainty in the data and the network's prediction, while preserving the strong modelling capabilities of DNNs [32]. In particular, BNNs can learn arbitrary data distributions from much simpler (e.g. Gaussian) weight distributions. This makes BNNs very appealing for robotic and medical applications [33] where uncertainty is a central component of data.

Despite the large body of literature on verifying safety of DNNs, the formal safety verification of BNNs has received less attention. Notably, [8, 45, 34] have proposed sampling-based techniques for obtaining probabilistic guarantees about BNNs. Although these approaches provide some insight

---

[*]Equal Contribution.

35th Conference on Neural Information Processing Systems (NeurIPS 2021).

into BNN safety, they suffer from two key limitations. First, sampling provides only bounds on the probability of the BNN's safety which is insufficient for systems with critical safety implications. For instance, having an autonomous vehicle with a $99.9\%$ safety guarantee is still insufficient for deployment if millions of vehicles are deployed. Second, samples can only simulate the system for a finite time, making it impossible to reason about the system's safety over an unbounded time horizon.

In this work, we study the safety verification problem for BNN policies in safety-critical systems over the infinite time horizon. Formally, we consider discrete-time closed-loop systems defined by a dynamical system and a BNN policy. Given a set of initial states and a set of unsafe (or bad) states, the goal of the safety verification problem is to verify that no system execution starting in an initial state can reach an unsafe state. Unlike existing literature which considers probability of safety, we verify *sure safety*, i.e. safety of every system execution of the system. In particular, we present a method for computing *safe weight sets* for which every system execution is safe as long as the BNN samples its weights from this set.

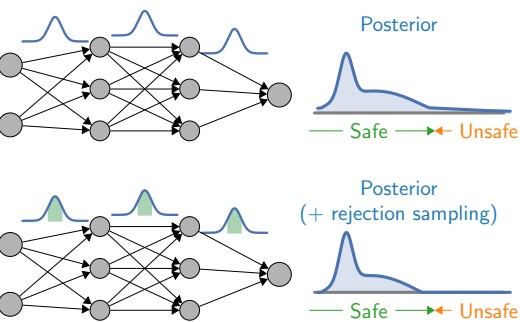

Figure 1: BNNs are typically unsafe by default. Top figure: The posterior of a typical BNN has unbounded support, resulting in a non-zero probability of producing an unsafe action. Bottom figure: Restricting the support of the weight distributions via rejection sampling ensures BNN safety.

Our approach to restrict the support of the weight distribution is necessary as BNNs with Gaussian weight priors typically produce output posteriors with unbounded support. Consequently, there is a low but non-zero probability for the output variable to lie in an unsafe region, see Figure 1. This implies that BNNs are usually unsafe by default. We therefore consider the more general problem of computing safe weight sets. Verifying that a weight set is safe allows re-calibrating the BNN policy by rejecting unsafe weight samples in order to guarantee safety.

As most BNNs employ uni-modal weight priors, e.g. Gaussians, we naturally adopt weight sets in the form of products of intervals centered at the means of the BNN's weight distributions. To verify safety of a weight set, we search for a safety certificate in the form of a *safe positive invariant* (also known as *safe inductive invariant*). A safe positive invariant is a set of system states that contains all initial states, is closed under the system dynamics and does not contain any unsafe state. The key advantage of using safe positive invariants is that their existence implies the *infinite time horizon safety*. We parametrize safe positive invariant candidates by (deterministic) neural networks that classify system states for determining set inclusion. Moreover, we phrase the search for an invariant as a learning problem. A separated verifier module then checks if a candidate is indeed a safe positive invariant by checking the required properties via constraint solving. In case the verifier finds a counterexample demonstrating that the candidate violates the safe positive invariant condition, we re-train the candidate on the found counterexample. We repeat this procedure until the verifier concludes that the candidate is a safe positive invariant ensuring that the system is safe.

The safe weight set obtained by our method can be used for safe exploration reinforcement learning. In particular, generating rollouts during learning by sampling from the safe weight set allows an exploration of the environment while ensuring safety. Moreover, projecting the (mean) weights onto the safe weight set after each gradient update further ensures that the improved policy stays safe.

**Contributions** Our contributions can be summarized as follows:

1. We define a safety verification problem for BNN policies which overcomes the unbounded posterior issue by computing and verifying safe weight sets. The problem generalizes the sure safety verification of BNNs and solving it allows re-calibrating BNN policies via rejection sampling to guarantee safety.
2. We introduce a method for computing safe weight sets in BNN policies in the form of products of intervals around the BNN weights' means. To verify safety of a weight set, our novel algorithm learns a safe positive invariant in the form of a deterministic neural network.
3. We evaluate our methodology on a series of benchmark applications, including non-linear systems and non-Lyapunovian safety specifications.

## 2 Related work

**Verification of feed-forward NN** Verification of robustness and safety properties in feed-forward DNNs has received much attention but remains an active research topic [26, 23, 17, 40, 7, 43]. As the majority of verification techniques were designed for deterministic NNs, they cannot be readily applied to BNNs. The safety verification of feed-forward BNNs has been considered in [8] by using samples to obtain statistical guarantees on the safety probability. The work of [45] also presents a sampling-based approach, however it provides certified lower bounds on the safety probability.

The literature discussed above considers NNs in isolation, which can provide input-output guarantees on a NN but are unable to reason holistically about the safety of the system that the NN is applied in. Verification methods that concern the safety of NNs interlinked with a system require different approaches than standalone NN verification, which we will discuss in the rest of this section.

**Finite time horizon safety of BNN policies** The work in [34] extends the method of [8] to verifying safety in closed-loop systems with BNN policies. However, similar to the standalone setting of [8], their method obtains only statistical guarantees on the safety probability and for the system's execution over a finite time horizon.

**Safe RL** Safe reinforcement learning has been primarily studied in the form of constrained Markov decision processes (CMDPs) [2, 18]. Compared to standard MDPs, an agent acting in a CMDP must satisfy an expected auxiliary cost term aggregated over an episode. The CMDP framework has been the base of several RL algorithms [44], notably the Constrained Policy Optimization (CPO) [1]. Despite these algorithms providing a decent performance, the key limitation of CMDPs is that the constraint is satisfied in expectation, which makes violations unlikely but nonetheless possible. Consequently, the CMDP framework is unsuited for systems where constraint violations are critical.

**Lyapunov-based stability** Safety in the context of "stability", i.e. always returning to a ground state, can be proved by Lyapunov functions [4]. Lyapunov functions have originally been considered to study stability of dynamical systems [29]. Intuitively, a Lyapunov function assigns a non-negative value to each state, and is required to decrease with respect to the system's dynamics at any state outside of the stable set. A Lyapunov-based method is proposed in [11] to ensure safety in CMDPs during training. Recently, the work of [9] presented a method for learning a policy as well as a neural network Lyapunov function which guarantees the stability of the policy. Similarly to our work, their learning procedure is counterexample-based. However, unlike [9], our work considers BNN policies and safety definitions that do not require returning to a set of ground states.

**Barrier functions for dynamical systems** Barrier functions can be used to prove infinite time horizon safety in dynamical systems [38, 39]. Recent works have considered learning neural network barrier functions [47], and a counterexample-based learning procedure is presented in [37].

**Finite time horizon safety of NN policies** Safety verification of continuous-time closed-loop systems with deterministic NN policies has been considered in [24, 19], which reduces safety verification to the reachability analysis in hybrid systems [10]. The work of [13] presents a method which computes a polynomial approximation of the NN policy to allow an efficient approximation of the reachable state set. Both works consider finite time horizon systems.

Our safety certificate most closely resembles inductive invariants for safety analysis in programs [14] and positive invariants for dynamical systems [5].

## 3 Preliminaries and problem statement

We consider a discrete-time dynamical system

$$\mathbf{x}_{t+1} = f(\mathbf{x}_t, \mathbf{u}_t), \ \mathbf{x}_0 \in \mathcal{X}_0.$$

The dynamics are defined by the function $f : \mathcal{X} \times \mathcal{U} \rightarrow \mathcal{X}$ where $\mathcal{X} \subseteq \mathbb{R}^m$ is the state space and $\mathcal{U} \subseteq \mathbb{R}^n$ is the control action space, $\mathcal{X}_0 \subseteq \mathcal{X}$ is the set of initial states and $t \in \mathbb{N}_{\geq 0}$ denotes a discretized time. At each time step $t$, the action is defined by the (possibly probabilistic) positional policy $\pi : \mathcal{X} \rightarrow \mathcal{D}(\mathcal{U})$, which maps the current state $\mathbf{x}_t$ to a distribution $\pi(\mathbf{x}_t) \in \mathcal{D}(U)$ over the set of actions. We use $\mathcal{D}(U)$ to denote the set of all probability distributions over $U$. The next action is then sampled according to $\mathbf{u}_t \sim \pi(\mathbf{x}_t)$, and together with the current state $\mathbf{x}_t$ of the system gives rise to the next state $\mathbf{x}_{t+1}$ of the system according to the dynamics $f$. Thus, the dynamics $f$ together

with the policy $\pi$ form a closed-loop system (or a feedback loop system). The aim of the policy is to maximize the expected cumulative reward (possibly discounted) from each starting state. Given a set of initial states $\mathcal{X}_0$ of the system, we say that a sequence of state-action pairs $(\mathbf{x}_t, \mathbf{u}_t)_{t=0}^{\infty}$ is a trajectory if $\mathbf{x}_0 \in \mathcal{X}_0$ and we have $\mathbf{u}_t \in \operatorname{supp}(\pi(\mathbf{x}_t))$ and $\mathbf{x}_{t+1} = f(\mathbf{x}_t, \mathbf{u}_t)$ for each $t \in \mathbb{N}_{\geq 0}$.

A neural network (NN) is a function $\pi : \mathbb{R}^m \to \mathbb{R}^n$ that consists of several sequentially composed layers $\pi = l_1 \circ \cdots \circ l_k$. Formally, a NN policy maps each system state to a Dirac-delta distribution which picks a single action with probability 1. Each layer $l_i$ is parametrized by learned weight values of the appropriate dimensions and an activation function $a$,

$$l_i(\mathbf{x}) = a(\mathbf{W}_i \mathbf{x} + \mathbf{b}_i), \mathbf{W}_i \in \mathbb{R}^{n_i \times m_i}, \mathbf{b}_i \in \mathbb{R}^{n_i}.$$

In this work, we consider ReLU activation functions $a(\mathbf{x}) = \operatorname{ReLU}(\mathbf{x}) = \max\{\mathbf{x}, \mathbf{0}\}$, although other piecewise linear activation such as the leaky-ReLU [25] and PReLU [21] are applicable as well.

In Bayesian neural networks (BNNs), weights are random variables and their values are sampled, each according to some distribution. Then each vector of sampled weights gives rise to a (deterministic) neural network. Given a training set $\mathcal{D}$, in order to train the BNN we assume a prior distribution $p(\mathbf{w}, \mathbf{b})$ over the weights. The learning then amounts to computing the posterior distribution $p(\mathbf{w}, \mathbf{b} \mid \mathcal{D})$ via the application of the Bayes rule. As analytical inference of the posterior is in general infeasible due to non-linearity introduced by the BNN architecture [32], practical training algorithms rely on approximate inference, e.g. Hamiltonian Monte Carlo [36], variational inference [6] or dropout [15].

When the policy in a dynamical system is a BNN, the policy maps each system state $\mathbf{x}_t$ to a probability distribution $\pi(\mathbf{x}_t)$ over the action space. Informally, this distribution is defined as follows. First, BNN weights $\mathbf{w}, \mathbf{b}$ are sampled according to the posterior BNN weight distribution, and the sampled weights give rise to a deterministic NN policy $\pi_{\mathbf{w}, \mathbf{b}}$. The action of the system is then defined as $\mathbf{u}_t = \pi_{\mathbf{w}, \mathbf{b}}(\mathbf{x}_t)$. Formal definition of the distribution $\pi(\mathbf{x}_t)$ is straightforward and proceeds by considering the product measure of distributions of all weights.

**Problem statement** We now define the two safety problems that we consider in this work. The first problem considers feed-forward BNNs, and the second problem considers closed-loop systems with BNN policies. While our solution to the first problem will be a subprocedure in our solution to the second problem, the reason why we state it as a separate problem is that we believe that our solution to the first problem is also of independent interest for the safety analysis of feed-forward BNNs.

Let $\pi$ be a BNN. Suppose that the vector $(\mathbf{w}, \mathbf{b})$ of BNN weights in $\pi$ has dimension $p + q$, where $p$ is the dimension of $\mathbf{w}$ and $q$ is the dimension of $\mathbf{b}$. For each $1 \leq i \leq p$, let $\mu_i$ denote the mean of the random variable $w_i$. Similarly, for each $1 \leq i \leq q$, let $\mu_{p+i}$ denote the mean of the random variable $b_i$. Then, for each $\epsilon \in [0, \infty]$, we define the set $W_\epsilon^\pi$ of weight vectors via

$$W_\epsilon^\pi = \prod_{i=1}^{p+q} [\mu_i - \epsilon, \mu_i + \epsilon] \subseteq \mathbb{R}^{p+q}.$$

We now proceed to defining our safety problem for feed-forward BNNs. Suppose that we are given a feed-forward BNN $\pi$, a set $\mathcal{X}_0 \subseteq \mathbb{R}^m$ of input points and a set $\mathcal{X}_u \subseteq \mathbb{R}^n$ of unsafe (or bad) output points. For a concrete vector $(\mathbf{w}, \mathbf{b})$ of weight values, let $\pi_{\mathbf{w}, \mathbf{b}}$ to be the (deterministic) NN defined by these weight values. We say that $\pi_{\mathbf{w}, \mathbf{b}}$ is safe if for each $\mathbf{x} \in \mathcal{X}_0$ we have $\pi_{\mathbf{w}, \mathbf{b}}(\mathbf{x}) \notin \mathcal{X}_u$, i.e. if evaluating $\pi_{\mathbf{w}, \mathbf{b}}$ on all input points does not lead to an unsafe output.

> **Problem 1** (Feed-forward BNNs). *Let $\pi$ be a feed-forward BNN, $\mathcal{X}_0 \subseteq \mathbb{R}^m$ a set of input points and $\mathcal{X}_u \subseteq \mathbb{R}^n$ a set of unsafe output points. Let $\epsilon \in [0, \infty]$. Determine whether each deterministic NN in $\{\pi_{\mathbf{w}, \mathbf{b}} \mid (\mathbf{w}, \mathbf{b}) \in W_\epsilon^\pi\}$ is safe.*

Next, we define our safety problem for closed-loop systems with BNN policies. Consider a closed-loop system defined by a dynamics function $f$, a BNN policy $\pi$ and an initial set of states $\mathcal{X}_0$. Let $\mathcal{X}_u \subseteq \mathcal{X}$ be a set of unsafe (or bad) states. We say that a trajectory $(\mathbf{x}_t, \mathbf{u}_t)_{t=0}^{\infty}$ is safe if $\mathbf{x}_t \notin \mathcal{X}_u$ for all $t \in \mathbb{N}_0$, hence if it does not reach any unsafe states. Note that this definition implies infinite time horizon safety of the trajectory. Given $\epsilon \in [0, \infty]$, define the set $\operatorname{Traj}_\epsilon^{f, \pi}$ to be the set of all system trajectories in which each sampled weight vector belongs to $W_\epsilon^\pi$.

**Problem 2** (Closed-loop systems with BNN policies). *Consider a closed-loop system defined by a dynamics function $f$, a BNN policy $\pi$ and a set of initial states $\mathcal{X}_0$. Let $\mathcal{X}_u$ be a set of unsafe states. Let $\epsilon \in [0, \infty]$. Determine whether each trajectory in $\mathsf{Traj}_\epsilon^{f,\pi}$ is safe.*

Note that the question of whether the BNN policy $\pi$ is safe (i.e. whether each trajectory of the system is safe) is a special case of the above problem which corresponds to $\epsilon = \infty$.

# 4 Main results

In this section we present our method for solving the safety problems defined in the previous section, with Section 4.1 considering Problem 1 and Section 4.2 considering Problem 2. Both problems consider safety verification with respect to a given value of $\epsilon \in [0, \infty]$, so in Section 4.3 we present our method for computing the value of $\epsilon$ for which our solutions to Problem 1 and Problem 2 may be used to verify safety. We then show in Section 4.4 how our new methodology can be adapted to the safe exploration RL setting.

## 4.1 Safe weight sets for feed-forward BNNs

Consider a feed-forward BNN $\pi$, a set $\mathcal{X}_0 \subseteq \mathbb{R}^m$ of inputs and a set $\mathcal{X}_u \subseteq \mathbb{R}^n$ of unsafe output of the BNN. Fix $\epsilon \in [0, \infty]$. To solve Problem 1, we show that the decision problem of whether each deterministic NN in $\{\pi_{\mathbf{w},\mathbf{b}} \mid (\mathbf{w}, \mathbf{b}) \in W_\epsilon^\pi\}$ is safe can be encoded as a system of constraints and reduced to constraint solving.

Suppose that $\pi = l_1 \circ \cdots \circ l_k$ consists of $k$ layers, with each $l_i(\mathbf{x}) = \mathrm{ReLU}(\mathbf{W}_i \mathbf{x} + \mathbf{b}_i)$. Denote by $\mathbf{M}_i$ the matrix of the same dimension as $\mathbf{W}_i$, with each entry equal to the mean of the corresponding random variable weight in $\mathbf{W}_i$. Similarly, define the vector $\mathbf{m}_i$ of means of random variables in $\mathbf{b}_i$. The real variables of our system of constraints are as follows, each of appropriate dimension:

- $\mathbf{x}_0$ encodes the BNN inputs, $\mathbf{x}_l$ encodes the BNN outputs;
- $\mathbf{x}_1^{\mathrm{in}}, \ldots, \mathbf{x}_{l-1}^{\mathrm{in}}$ encode vectors of input values of each neuron in the hidden layers;
- $\mathbf{x}_1^{\mathrm{out}}, \ldots, \mathbf{x}_{l-1}^{\mathrm{out}}$ encode vectors of output values of each neuron in the hidden layers;
- $\mathbf{x}_{0,\mathrm{pos}}$ and $\mathbf{x}_{0,\mathrm{neg}}$ are dummy variable vectors of the same dimension as $\mathbf{x}_0$ and which will be used to distinguish between positive and negative NN inputs in $\mathbf{x}_0$, respectively.

We use $\mathbf{1}$ to denote the vector/matrix whose all entries are equal to 1, of appropriate dimensions defined by the formula in which it appears. Our system of constraints is as follows:

$$\mathbf{x}_0 \in \mathcal{X}_0, \quad \mathbf{x}_l \in \mathcal{X}_u \qquad \text{(Input-output conditions)}$$

$$\mathbf{x}_i^{\mathrm{out}} = \mathrm{ReLU}(\mathbf{x}_i^{\mathrm{in}}), \text{ for each } 1 \le i \le l-1 \qquad \text{(ReLU encoding)}$$

$$
\begin{aligned}
(\mathbf{M}_i - \epsilon \cdot \mathbf{1})\mathbf{x}_i^{\mathrm{out}} + (\mathbf{m}_i - \epsilon \cdot \mathbf{1}) &\le \mathbf{x}_{i+1}^{\mathrm{in}}, \text{ for each } 1 \le i \le l-1 \\
\mathbf{x}_{i+1}^{\mathrm{in}} &\le (\mathbf{M}_i + \epsilon \cdot \mathbf{1})\mathbf{x}_i^{\mathrm{out}} + (\mathbf{m}_i + \epsilon \cdot \mathbf{1}), \text{ for each } 1 \le i \le l-1
\end{aligned}
\qquad \text{(BNN hidden layers)}
$$

$$
\begin{aligned}
\mathbf{x}_{0,\mathrm{pos}} &= \mathrm{ReLU}(\mathbf{x}_0), \quad \mathbf{x}_{0,\mathrm{neg}} = -\mathrm{ReLU}(-\mathbf{x}_0) \\
(\mathbf{M}_0 - \epsilon \cdot \mathbf{1})\mathbf{x}_{0,\mathrm{pos}} + (\mathbf{M}_0 + \epsilon \cdot \mathbf{1})\mathbf{x}_{0,\mathrm{neg}} + (\mathbf{m}_0 - \epsilon \cdot \mathbf{1}) &\le \mathbf{x}_1^{\mathrm{in}} \\
\mathbf{x}_1^{\mathrm{in}} &\le (\mathbf{M}_0 + \epsilon \cdot \mathbf{1})\mathbf{x}_0^{\mathrm{out}} + (\mathbf{M}_0 - \epsilon \cdot \mathbf{1})\mathbf{x}_{0,\mathrm{neg}} + (\mathbf{m}_0 + \epsilon \cdot \mathbf{1})
\end{aligned}
\qquad \text{(BNN input layer)}
$$

Denote by $\Phi(\pi, \mathcal{X}_0, \mathcal{X}_u, \epsilon)$ the system of constraints defined above. The proof of Theorem 1 shows that it encodes that $\mathbf{x}_0 \in \mathcal{X}_0$ is an input point for which the corresponding output point of $\pi$ is unsafe, i.e. $\mathbf{x}_l = \pi(\mathbf{x}_0) \in \mathcal{X}_u$. The first equation encodes the input and output conditions. The second equation encodes the ReLU input-output relation for each hidden layer neuron. The remaining equations encode the relation between neuron values in successive layers of the BNN as well as that the sampled BNN weight vector is in $W_\epsilon^\pi$. For hidden layers, we know that the output value of each neuron is nonnegative, i.e. $\mathbf{x}_i^{\mathrm{out}} \ge \mathbf{0}$ for the $i$-th hidden layer where $1 \le i \le l-1$, and so

$$(\mathbf{M}_i - \epsilon \cdot \mathbf{1})\mathbf{x}_i^{\mathrm{out}} \le (\mathbf{M}_i + \epsilon \cdot \mathbf{1})\mathbf{x}_i^{\mathrm{out}}.$$

Hence, the BNN weight relation with neurons in the successive layer as well as the fact that the sampled weights are in $W_\epsilon^\pi$ is encoded as in equations 3-4 above. For the input layer, however, we do not know the signs of the input neuron values $\mathbf{x}_0$ and so we introduce dummy variables $\mathbf{x}_{0,\text{pos}} = \text{ReLU}(\mathbf{x}_0)$ and $\mathbf{x}_{0,\text{neg}} = -\text{ReLU}(-\mathbf{x}_0)$ in equation 5. This allows encoding the BNN weight relations between the input layer and the first hidden layer as well as the fact that the sampled weight vector is in $W_\epsilon^\pi$, as in equations 6-7. Theorem 1 shows that Problem 1 is equivalent to solving the system of constraints $\Phi(\pi, \mathcal{X}_0, \mathcal{X}_u, \epsilon)$. Its proof can be found in the Supplementary Material.

**Theorem 1.** *Let $\epsilon \in [0, \infty]$. Then each deterministic NN in $\{\pi_{\mathbf{w},\mathbf{b}} \mid (\mathbf{w}, \mathbf{b}) \in W_\epsilon^\pi\}$ is safe if and only if the system of constraints $\Phi(\pi, \mathcal{X}_0, \mathcal{X}_u, \epsilon)$ is not satisfiable.*

**Solving the constraints** Observe that $\epsilon$, $\mathbf{M}_i$ and $\mathbf{m}_i$, $1 \le i \le l-1$, are constant values that are known at the time of constraint encoding. Thus, in $\Phi(\pi, \mathcal{X}_0, \mathcal{X}_u, \epsilon)$, only the ReLU constraints and possibly the input-output conditions are not linear. Depending on the form of $\mathcal{X}_0$ and $\mathcal{X}_u$ and on how we encode the ReLU constraints, we may solve the system $\Phi(\pi, \mathcal{X}_0, \mathcal{X}_u, \epsilon)$ in several ways:

1. **MILP.** It is shown in [31, 12, 43] that the ReLU relation between two real variables can be encoded via mixed-integer linear constraints (MILP) by introducing 0/1-integer variables to encode whether a given neuron is active or inactive. Hence, if $\mathcal{X}_0$ and $\mathcal{X}_u$ are given by linear constraints, we may solve $\Phi(\pi, \mathcal{X}_0, \mathcal{X}_u, \epsilon)$ by a MILP solver. The ReLU encoding requires that each neuron value is bounded, which is ensured if $\mathcal{X}_0$ is a bounded set and if $\epsilon < \infty$.
2. **Reluplex.** In order to allow unbounded $\mathcal{X}_0$ and $\epsilon = \infty$, we may use algorithms based on the Reluplex calculus [26, 27] to solve $\Phi(\pi, \mathcal{X}_0, \mathcal{X}_u, \epsilon)$. Reluplex is an extension of the standard simplex algorithm for solving systems of linear constraints, designed to allow ReLU constraints as well. While Reluplex does not impose the boundedness condition, it is in general less scalable than MILP-solving.
3. **NRA-SMT.** Alternatively, if $\mathcal{X}_0$ or $\mathcal{X}_u$ are given by non-linear constraints we may solve them by using an NRA-SMT-solver (non-linear real arithmetic satisfiability modulo theory), e.g. dReal [16]. To use an NRA-SMT-solver, we can replace the integer 0/1-variables of the ReLU neuron relations encoding with real variables that satisfy the constraint $x(x-1) = 0$. While NRA-SMT is less scalable compared to MILP, we note that it has been used in previous works on RL stability verification [9].

**Safety via rejection sampling** As discussed in Section 1, once the safety of NNs in $\{\pi_{\mathbf{w},\mathbf{b}} \mid (\mathbf{w}, \mathbf{b}) \in W_\epsilon^\pi\}$ has been verified, we can "re-calibrate" the BNN to reject sampled weights which are not in $W_\epsilon^\pi$. Hence, rejection sampling gives rise to a safe BNN.

## 4.2 Safe weight sets for closed-loop systems with BNN Policies

Now consider a closed-loop system with a dynamics function $f : \mathcal{X} \times \mathcal{U} \to \mathcal{X}$ with $\mathcal{X} \subseteq \mathbb{R}^m$ and $\mathcal{U} \subseteq \mathbb{R}^n$, a BNN policy $\pi$, an initial state set $\mathcal{X}_0 \subseteq \mathcal{X}$ and an unsafe state set $\mathcal{X}_u \subseteq \mathcal{X}$. Fix $\epsilon \in [0, \infty]$. In order to solve Problem 2 and verify safety of each trajectory contained in $\text{Traj}_\epsilon^{f,\pi}$, our method searches for a positive invariant-like safety certificate that we define below.

**Positive invariants for safety** A positive invariant in a dynamical system is a set of states which contains all initial states and which is closed under the system dynamics. These conditions ensure that states of all system trajectories are contained in the positive invariant. Hence, a positive invariant which does not contain any unsafe states can be used to certify safety of every trajectory over infinite time horizon. In this work, however, we are not trying to prove safety of every trajectory, but only of those trajectories contained in $\text{Traj}_\epsilon^{f,\pi}$. To that end, we define $W_\epsilon^\pi$-safe positive invariants. Intuitively, a $W_\epsilon^\pi$-safe positive invariant is required to contain all initial states, to be closed under the dynamics $f$ and the BNN policy $\pi$ when the sampled weight vector is in $W_\epsilon^\pi$, and not to contain any unsafe state.

**Definition 1** ($W_\epsilon^\pi$-safe positive invariants). *A set $\text{Inv} \subseteq \mathcal{X}$ is said to be a $W_\epsilon^\pi$-positive invariant if $\mathcal{X}_0 \subseteq \text{Inv}$, for each $\mathbf{x} \in \text{Inv}$ and $(\mathbf{w}, \mathbf{b}) \in W_\epsilon^\pi$ we have that $f(\mathbf{x}, \pi_{\mathbf{w},\mathbf{b}}(\mathbf{x})) \in \text{Inv}$, and $\text{Inv} \cap \mathcal{X}_u = \emptyset$.*

Theorem 2 shows that $W_\epsilon^\pi$-safe positive invariants can be used to verify safety of all trajectories in $\text{Traj}_\epsilon^{f,\pi}$ in Problem 2. The proof is straightforward and is deferred to the Supplementary Material.

**Theorem 2.** *If there exists a $W_\epsilon^\pi$-safe positive invariant, then each trajectory in $\text{Traj}_\epsilon^{f,\pi}$ is safe.*

**Algorithm 1** Learning algorithm for $W_\epsilon^\pi$-safe positive invariants

---

**Input** Dynamics function $f$, BNN policy $\pi$, Initial state set $\mathcal{X}_0$, Unsafe state set $\mathcal{X}_u$, $\epsilon \in [0, \infty]$
$\tilde{\mathcal{X}}_0, \tilde{\mathcal{X}}_u \leftarrow$ random samples of $\mathcal{X}_0, \mathcal{X}_u$
$D_{\text{spec}} \leftarrow \tilde{\mathcal{X}}_u \times \{0\} \cup \tilde{\mathcal{X}}_0 \times \{1\}$, $D_{\text{ce}} \leftarrow \{\}$
Optional (bootstrapping): $S_{\text{bootstrap}} \leftarrow$ sample finite trajectories with initial state sampled from $\mathcal{X}$
        $D_{\text{spec}} \leftarrow D_{\text{spec}} \cup \{(\mathbf{x}, 0) | \exists s \in S_{\text{bootstrap}} \text{ that starts in } \mathbf{x} \in \mathcal{X} \text{ and reaches } \mathcal{X}_u\}$
        $\cup \{(\mathbf{x}, 1) | \exists s \in S_{\text{bootstrap}} \text{ that starts in } \mathbf{x} \in \mathcal{X} \text{ and does not reach } \mathcal{X}_u\}$
Pre-train neural network $g^{\text{Inv}}$ on datasets $D_{\text{spec}}$ and $D_{\text{ce}}$ with loss function $\mathcal{L}$
**while** timeout not reached **do**
    **if** $\exists(\mathbf{x}, \mathbf{x}', \mathbf{u}, \mathbf{w}, \mathbf{b})$ s.t. $g^{\text{Inv}}(\mathbf{x}) \geq 0$, $g^{\text{Inv}}(\mathbf{x}') < 0$, $(\mathbf{w}, \mathbf{b}) \in W_\epsilon^\pi$, $\mathbf{u} = \pi_{\mathbf{w}, \mathbf{b}}(\mathbf{x})$, $\mathbf{x}' = f(\mathbf{x}, \mathbf{u})$
    **then**
        $D_{\text{ce}} \leftarrow D_{\text{ce}} \cup \{(\mathbf{x}, \mathbf{x}')\}$
    **else if** $\exists(\mathbf{x})$ s.t. $\mathbf{x} \in \mathcal{X}_0$, $g^{\text{Inv}}(\mathbf{x}) < 0$ **then**
        $D_{\text{spec}} \leftarrow D_{\text{spec}} \cup \{(\mathbf{x}, 1)\}$
    **else if** $\exists(\mathbf{x})$ s.t. $\mathbf{x} \in \mathcal{X}_u$, $g^{\text{Inv}}(\mathbf{x}) \geq 0$ **then**
        $D_{\text{spec}} \leftarrow D_{\text{spec}} \cup \{(\mathbf{x}, 0)\}$
    **else**
        **Return** Safe
    **end if**
    Train neural network $g^{\text{Inv}}$ on datasets $D_{\text{spec}}$ and $D_{\text{ce}}$ with loss function $\mathcal{L}$
**end while**
**Return** Unsafe

---

**Learning positive invariants** We now present a learning algorithm for a $W_\epsilon^\pi$-safe positive invariant. It learns a neural network $g^{\text{Inv}} : \mathbb{R}^m \to \mathbb{R}$, where the positive invariant is then defined as the set $\text{Inv} = \{\mathbf{x} \in \mathcal{X} \mid g^{\text{Inv}}(\mathbf{x}) \geq 0\}$. The pseudocode is given in Algorithm 1.

The algorithm first samples $\tilde{\mathcal{X}}_0$ from $\mathcal{X}_0$ and $\tilde{\mathcal{X}}_u$ from $\mathcal{X}_u$ and initializes the specification set $D_{\text{spec}}$ to $\tilde{\mathcal{X}}_u \times \{0\} \cup \tilde{\mathcal{X}}_0 \times \{1\}$ and the counterexample set $D_{\text{ce}}$ to an empty set. Optionally, the algorithm also bootstraps the positive invariant network by initializing $D_{\text{spec}}$ with random samples from the state space $\mathcal{X}$ labeled with Monte-Carlo estimates of reaching the unsafe states. The rest of the algorithm consists of two modules which are composed into a loop: the *learner* and the *verifier*. In each loop iteration, the learner first learns a $W_\epsilon^\pi$-safe positive invariant candidate which takes the form of a neural network $g^{\text{Inv}}$. This is done by minimizing the loss function $\mathcal{L}$ that depends on $D_{\text{spec}}$ and $D_{\text{ce}}$:

$$\mathcal{L}(g^{\text{Inv}}) = \frac{1}{|D_{\text{spec}}|} \sum_{(\mathbf{x}, y) \in D_{\text{spec}}} \mathcal{L}_{\text{cls}}\big(g^{\text{Inv}}(\mathbf{x}), y\big) + \lambda \frac{1}{|D_{\text{ce}}|} \sum_{(\mathbf{x}, \mathbf{x}') \in D_{\text{ce}}} \mathcal{L}_{\text{ce}}\big(g^{\text{Inv}}(\mathbf{x}), g^{\text{Inv}}(\mathbf{x}')\big), \quad (1)$$

where $\lambda$ is a tuning parameter and $\mathcal{L}_{\text{cls}}$ a binary classification loss function, e.g. the 0/1-loss $\mathcal{L}_{0/1}(z, y) = \mathbb{1}[\mathbb{1}[z \geq 0] \neq y]$ or the logistic loss $\mathcal{L}_{\log}(z, y) = z - z \cdot y + \log(1 + \exp(-z))$ as its differentiable alternative. The term $\mathcal{L}_{\text{ce}}$ is the counterexample loss which we define via

$$\mathcal{L}_{\text{ce}}(z, z') = \mathbb{1}\big[z > 0\big]\mathbb{1}\big[z' < 0\big]\mathcal{L}_{\text{cls}}\big(z, 0\big)\mathcal{L}_{\text{cls}}\big(z', 1\big). \quad (2)$$

Intuitively, the first sum in eq. (1) forces $g^{\text{Inv}}$ to be nonnegative at initial states and negative at unsafe states contained in $D_{\text{spec}}$, and the second term forces each counterexample in $D_{\text{ce}}$ not to destroy the closedness of $\text{Inv}$ under the system dynamics.

Once $g^{\text{Inv}}$ is learned, the verifier checks whether $\text{Inv}$ is indeed a $W_\epsilon^\pi$-safe positive invariant. To do this, the verifier needs to check the three defining properties of $W_\epsilon^\pi$-safe positive invariants:

1. *Closedness of* $\text{Inv}$ *under system dynamics.* The verifier checks if there exist states $\mathbf{x} \in \text{Inv}$, $\mathbf{x}' \notin \text{Inv}$ and a BNN weight vector $(\mathbf{w}, \mathbf{b}) \in W_\epsilon^\pi$ such that $f(\mathbf{x}, \pi_{\mathbf{w}, \mathbf{b}}(\mathbf{x})) = \mathbf{x}'$. To do this, it introduces real variables $\mathbf{x}, \mathbf{x}' \in \mathbb{R}^m$, $\mathbf{u} \in \mathbb{R}^n$ and $y, y' \in \mathbb{R}$, and solves:

        maximize $y - y'$ subject to

        $y \geq 0, y' < 0, y = g^{\text{Inv}}(\mathbf{x}), y' = g^{\text{Inv}}(\mathbf{x}')$

        $\mathbf{x}' = f(\mathbf{x}, \mathbf{u})$

        $\mathbf{u}$ is an output of $\pi$ on input $\mathbf{x}$ and weights $(\mathbf{w}, \mathbf{b}) \in W_\epsilon^\pi$

The conditions $y = g^{\mathsf{Inv}}(\mathbf{x})$ and $y' = g^{\mathsf{Inv}}(\mathbf{x}')$ are encoded by using the existing techniques for encoding deterministic NNs as systems of MILP/Reluplex/NRA-SMT constraints. The condition in the third equation is encoded simply by plugging variable vectors $\mathbf{x}$ and $\mathbf{u}$ into the equation for $f$. Finally, for condition in the fourth equation we use our encoding from Section 4.1 where we only need to omit the Input-output condition. The optimization objective is added in order to search for the "worst" counterexample. We note that MILP [20] and SMT [16] solvers allow optimizing linear objectives, and recently Reluplex algorithm [26] has also been extended to allow solving optimization problems [42]. If a counterexample $(\mathbf{x}, \mathbf{x}')$ is found, it is added to $D_{\mathrm{ce}}$ and the learner tries to learn a new candidate. If the system of constraints is unsatisfiable, the verifier proceeds to the second check.

2. *Non-negativity on $\mathcal{X}_0$.* The verifier checks if there exists $\mathbf{x} \in \mathcal{X}_0$ for which $g^{\mathsf{Inv}}(\mathbf{x}) < 0$. If such $\mathbf{x}$ is found, $(\mathbf{x}, 1)$ is added to $D_{\mathrm{spec}}$ and the learner then tries to learn a new candidate. If the system of constraints is unsatisfiable, the verifier proceeds to the third check.

3. *Negativity on $\mathcal{X}_u$.* The verifier checks if there exists $\mathbf{x} \in \mathcal{X}_u$ with $g^{\mathsf{Inv}}(\mathbf{x}) \geq 0$. If such $\mathbf{x}$ is found, $(\mathbf{x}, 0)$ is added to $D_{\mathrm{spec}}$ and the learner then tries to learn a new candidate. If the system of constraints is unsatisfiable, the veririfer concludes that Inv is a $W_\epsilon^\pi$-positive invariant which does not contain any unsafe state and so each trajectory in $\mathsf{Traj}_\epsilon^{f,\pi}$ is safe.

Theorem 3 shows that neural networks $f^{\mathsf{Inv}}$ for which Inv is a $W_\epsilon^\pi$-safe positive invariants are global minimizers of the loss function $\mathcal{L}$ with the 0/1-classification loss. Theorem 4 establishes the correctness of our algorithm. Proofs can be found in the Supplementary Material.

**Theorem 3.** *The loss function $\mathcal{L}$ is nonnegative for any neural network $g$, i.e. $\mathcal{L}(g) \geq 0$. Moreover, if* Inv *is a $W_\epsilon^\pi$-safe positive invariant and $\mathcal{L}_{cls}$ the 0/1-loss, then $\mathcal{L}(g^{\mathsf{Inv}}) = 0$. Hence, neural networks $g^{\mathsf{Inv}}$ for which* Inv *is a $W_\epsilon^\pi$-safe positive invariant are global minimizers of the loss function $\mathcal{L}$ when $\mathcal{L}_{cls}$ is the 0/1-loss.*

**Theorem 4.** *If the verifier in Algorithm 1 shows that constraints in three checks are unsatisfiable, then the computed* Inv *is indeed a $W_\epsilon^\pi$-safe positive invariant. Hence, Algorithm 1 is correct.*

**Safety via rejection sampling** As discussed in Section 1, once the safety of all trajectories in $\mathsf{Traj}_\epsilon^{f,\pi}$ has been verified, we can "re-calibrate" the BNN policy to reject sampled weights which are not in $W_\epsilon^\pi$. Hence, rejection sampling gives rise to a safe BNN policy.

### 4.3 Computation of safe weight sets and the value of $\epsilon$

Problems 1 and 2 assume a given value of $\epsilon$ for which safety needs to be verified. In order to compute the largest value of $\epsilon$ for which our approach can verify safety, we start with a small value of $\epsilon$ and iteratively increase it until we reach a value that cannot be certified or until the timeout is reached, in order to compute as large safe weight set as possible. In each iteration, our Algorithm 1 does not start from scratch but is initialized with the $g^{\mathsf{Inv}}$ and $D_{\mathrm{spec}}$ from the previous successful iteration, i.e. attempting to enlarge the current safe weight set. Our iterative process significantly speeds up the search process compared to naively restarting our algorithm in every iteration.

### 4.4 Safe exploration reinforcement learning

Given a safe but non-optimal initial policy $\pi_0$, safe exploration reinforcement learning (SERL) concerns the problem of improving the expected return of $\pi_0$ while ensuring safety when collecting samples of the environment [44, 1, 35]. Our method from Section 4.2 for computing safe weight sets can be adapted to this setting with minimal effort. In particular, the safety bound $\epsilon$ for the intervals centered at the weight means can be used in combination with the rejection sampling to generate safe but randomized rollouts on the environment. Moreover, $\epsilon$ provides bounds on the gradient updates when optimizing the policy using Deep Q-learning or policy gradient methods, i.e., performing *projected gradient descent*. We sketch an algorithm for SERL in the Supplementary Material.

## 5 Experiments

We perform an experimental evaluation of our proposed method for learning positive invariant neural networks that prove infinite time horizon safety. Our evaluation consists of an ablation study where we disable different core components of Algorithm 1 and measure their effects on the obtained

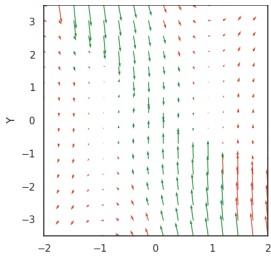

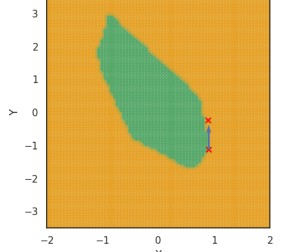

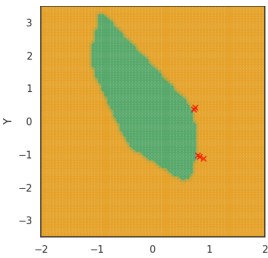

(a) Vector field of the system when controlled by the posterior's mean. Green/red arrows indicate empirical safe/unsafe estimates.

(b) First guess of $g^{\mathsf{Inv}}$. Green area shows $g^{\mathsf{Inv}} > 0$, orange area $g^{\mathsf{Inv}} < 0$. Red markers show the found counterexample.

(c) Final $g^{\mathsf{Inv}}$ proving the safety of the system. Previous counterexamples marked in red.

Figure 2: Invariant learning shown on the inverted pendulum benchmark.

safety bounds and the algorithm's runtime. First, we run the algorithm without any re-training on the counterexamples. In the second step, we run Algorithm 1 by initializing $D_{\text{spec}}$ with samples from $\mathcal{X}_0$ and $\mathcal{X}_u$ only. Finally, we bootstrap the positive invariant network by initializing $D_{\text{spec}}$ with random samples from the state space labeled with Monte-Carlo estimates of reaching the unsafe states. We consider environments with a piecewise linear dynamic function, initial and unsafe state sets so that the verification steps of our algorithm can be reduced to MILP-solving using Gurobi [20]. Details on our evaluation are in the Supplementary Material. Code is publicly available [2].

We conduct our evaluation on three benchmark environments that differ in terms of complexity and safety specifications. We train two BNN policies for each benchmark-ablation pair, one with Bayesian weights from the second layer on (with $\mathcal{N}(0, 0.1)$ prior) and one with Bayesian weights in all layers (with $\mathcal{N}(0, 0.05)$ prior). Recall, in our BNN encoding in Section 4.1, we showed that encoding of the BNN input layer requires additional constraints and extra care, since we do not know the signs of input neuron values. Hence, we consider two BNN policies in our evaluation in order to study how the encoding of the input layer affects the safe weight set computation.

Our first benchmark represents an unstable linear dynamical system of the form $x_{t+1} = Ax_t + Bu_t$. A BNN policy stabilizes the system towards the point $(0, 0)$. Consequently, the set of unsafe states is defined as $\{x \in \mathbb{R}^2 \mid |x|_\infty \geq 1.2\}$, and the initial states as $\{x \in \mathbb{R}^2 \mid |x|_\infty \leq 0.6\}$.

Our second benchmark is the inverted pendulum task, which is a classical non-linear control problem. The two state variables $a$ and $b$ represent the angle and angular velocity of a pendulum that must be controlled in an upward direction. The actions produced by the policy correspond to a torque applied to the anchor point. Our benchmark concerns a variant of the original problem where the non-linearity in $f$ is expressed by piecewise linear functions. The resulting system, even with a trained policy, is highly unstable, as shown in Figure 2. The set of initial states corresponds to pendulum states in an almost upright position and with small angular velocity. The set of unsafe states represents the pendulum falling down. Figure 2 visualizes the system and the learned invariant's decision boundary for the inverted pendulum task.

While the previous two benchmarks concern stability specifications, we evaluate our method on a non-Lyapunovian safety specification in our third benchmark. In particular, our third benchmark is a collision avoidance task, where the system does not stabilize to the same set of terminal states in every execution. The system is described by three variables. The first variable specifies the agent's own vertical location, while the other two variables specify an intruder's vertical and horizontal position. The objective is to avoid colliding with the intruder who is moving toward the agent by lateral movement commands as the policy's actions. The initial states represent far-away intruders, and crashes with the intruder define the unsafe states.

Table 1 shows the results of our evaluation. Our results demonstrate that re-training with the counterexamples is the key component that determines our algorithm's success. In all cases, except for the linear dynamical system, the initial guess of the invariant candidate violates the invariant

---

[2]`https://github.com/mlech26l/bayesian_nn_safety`

Table 1: Results of our benchmark evaluation. The epsilon values are multiples of the weight's standard deviation $\sigma$. We evaluated several epsilon values, and the table shows the largest that could be proven safe. A dash "-" indicates an unsuccessful invariant search. Runtime in seconds.

| Environment | No re-training | | Init $D_{spec}$ with $\mathcal{X}_0$ and $\mathcal{X}_u$ | | Bootstrapping $D_{spec}$ | |
|---|---|---|---|---|---|---|
| | Verified | Runtime | Verified | Runtime | Verified | Runtime |
| Unstable LDS | - | 3 | $1.5\sigma$ | 569 | $2\sigma$ | 760 |
| Unstable LDS (all) | $0.2\sigma$ | 3 | $0.5\sigma$ | 6 | $0.5\sigma$ | 96 |
| Pendulum | - | 2 | $2\sigma$ | 220 | $2\sigma$ | 40 |
| Pendulum (all) | - | 2 | $0.2\sigma$ | 1729 | $1.5\sigma$ | 877 |
| Collision avoid. | - | 2 | - | - | $2\sigma$ | 154 |
| Collision avoid. (all) | - | 2 | - | - | $1.5\sigma$ | 225 |

condition. Moreover, boostrapping $D_{\text{spec}}$ with random points labeled by empirical estimates of reaching the unsafe states improves the search process significantly.

## 6  Conclusion

In this work we formulated the safety verification problem for BNN policies in infinite time horizon systems, that asks to compute safe BNN weight sets for which every system execution is safe as long as the BNN samples its weights from this set. Solving this problem allows re-calibrating the BNN policy to reject unsafe weight samples in order to guarantee system safety. We then introduced a methodology for computing safe weight sets in BNN policies in the form of products of intervals around the BNN weight's means, and a method for verifying their safety by learning a positive invariant-like safety certificate. We believe that our results present an important first step in guaranteeing safety of BNNs for deployment in safety-critical scenarios. While adopting products of intervals around the BNN's weight means is a natural choice given that BNN priors are typically unimodal distributions, this is still a somewhat restrictive shape for safe weight sets. Thus, an interesting direction of future work would be to study more general forms of safe weight sets that could be used for re-calibration of BNN posteriors and their safety verification. Another interesting problem would be to design an approach for refuting a weight set as unsafe which would complement our method, or to consider closed-loop systems with stochastic environment dynamics.

Any verification method for neural networks, even more so for neural networks in feedback loops, suffers from scalability limitations due to the underlying complexity class [26, 24]. Promising research directions on improving the scalability of our approach by potentially speeding up the constraint solving step are gradient based optimization techniques [22] and to incorporate the constraint solving step already in the training procedure [46].

Since the aim of AI safety is to ensure that systems do not behave in undesirable ways and that safety violating events are avoided, we are not aware of any potential negative societal impacts.

## Acknowledgments

This research was supported in part by the Austrian Science Fund (FWF) under grant Z211-N23 (Wittgenstein Award), ERC CoG 863818 (FoRM-SMArt), and the European Union's Horizon 2020 research and innovation programme under the Marie Skłodowska-Curie Grant Agreement No. 665385.

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
