# Supplementary Material

**Mathias Lechner**[*]
IST Austria
Klosterneuburg, Austria
mlechner@ist.ac.at

**Đorđe Žikelić**[*]
IST Austria
Klosterneuburg, Austria
dzikelic@ist.ac.at

**Krishnendu Chatterjee**
IST Austria
Klosterneuburg, Austria
kchatterjee@ist.ac.at

**Thomas A. Henzinger**
IST Austria
Klosterneuburg, Austria
tah@ist.ac.at

## A  Proofs

**Theorem 1.** *Let $\epsilon \in [0, \infty]$. Then each deterministic NN in $\{\pi_{\mathbf{w}, \mathbf{b}} \mid (\mathbf{w}, \mathbf{b}) \in W_\epsilon^\pi\}$ is safe if and only if the system of constraints $\Phi(\pi, \mathcal{X}_0, \mathcal{X}_u, \epsilon)$ is not satisfiable.*

*Proof.* We prove the equivalent claim that there exists a weight vector $(\mathbf{w}, \mathbf{b}) \in W_\epsilon^\pi$ for which $\pi_{\mathbf{w}, \mathbf{b}}$ is unsafe if and only if $\Phi(\pi, \mathcal{X}_0, \mathcal{X}_u, \epsilon)$ is satisfiable.

First, suppose that there exists a weight vector $(\mathbf{w}, \mathbf{b}) \in W_\epsilon^\pi$ for which $\pi_{\mathbf{w}, \mathbf{b}}$ is unsafe and we want to show that $\Phi(\pi, \mathcal{X}_0, \mathcal{X}_u, \epsilon)$ is satisfiable. This direction of the proof is straightforward since values of the network's neurons on the unsafe input give rise to a solution of $\Phi(\pi, \mathcal{X}_0, \mathcal{X}_u, \epsilon)$. Indeed, by assumption there exists a vector of input neuron values $\mathbf{x}_0 \in \mathcal{X}_0$ for which the corresponding vector of output neuron values $\mathbf{x}_l = \pi_{\mathbf{w}, \mathbf{b}}(\mathbf{x}_0)$ is unsafe, i.e. $\mathbf{x}_l \in \mathcal{X}_u$. By defining $\mathbf{x}_i^{\text{in}}, \mathbf{x}_i^{\text{out}}$ to be the vectors of the corresponding input and output neuron values for the $i$-th hidden layer for each $1 \leq i \leq l-1$ and by setting $\mathbf{x}_{0,\text{pos}} = \text{ReLU}(\mathbf{x_0})$ and $\mathbf{x}_{0,\text{neg}} = -\text{ReLU}(-\mathbf{x_0})$, we easily see that these variable values satisfy the Input-output conditions, the ReLU encoding conditions and the BNN input and hidden layer conditions, therefore we get a solution to the system of constraints $\Phi(\pi, \mathcal{X}_0, \mathcal{X}_u, \epsilon)$.

We now proceed to the more involved direction of this proof and show that any solution to the system of constraints $\Phi(\pi, \mathcal{X}_0, \mathcal{X}_u, \epsilon)$ gives rise to weights $(\mathbf{w}, \mathbf{b}) \in W_\epsilon^\pi$ for which $\pi_{\mathbf{w}, \mathbf{b}}$ is unsafe. Let $\mathbf{x}_0$, $\mathbf{x}_l$, $\mathbf{x}_{0,\text{pos}}$, $\mathbf{x}_{0,\text{neg}}$ and $\mathbf{x}_i^{\text{in}}, \mathbf{x}_i^{\text{out}}$ for $1 \leq i \leq l-1$, be real vectors that satisfy the system of constraints $\Phi(\pi, \mathcal{X}_0, \mathcal{X}_u, \epsilon)$. Fix $1 \leq i \leq l-1$. From the BNN hidden layers constraint for layer i, we have

$$(\mathbf{M}_i - \epsilon \cdot \mathbf{1})\mathbf{x}_i^{\text{out}} + (\mathbf{m}_i - \epsilon \cdot \mathbf{1}) \leq \mathbf{x}_{i+1}^{\text{in}} \leq (\mathbf{M}_i + \epsilon \cdot \mathbf{1})\mathbf{x}_i^{\text{out}} + (\mathbf{m}_i + \epsilon \cdot \mathbf{1}). \tag{1}$$

We show that there exist values $\mathbf{W}_i^*, \mathbf{b}_i^*$ of BNN weights between layers $i$ and $i+1$ such that each weight value is at most $\epsilon$ apart from its mean, and such that $\mathbf{x}_{i+1}^{\text{in}} = \mathbf{W}_i^* \mathbf{x}_i^{\text{out}} + \mathbf{b}_i^*$. Indeed, to formally show this, we define $W_\epsilon^\pi[i]$ to be the set of all weight vectors between layers $i$ and $i+1$ such that each weight value is distant from its mean by at most $\epsilon$ (hence, $W_\epsilon^\pi[i]$ is a projection of $W_\epsilon^\pi$ onto dimensions that correspond to the weights between layers $i$ and $i+1$). We then consider a continuous function $h_i : W_\epsilon^\pi[i] \to \mathbb{R}$ defined via

$$h_i(\mathbf{W}_i, \mathbf{b}_i) = \mathbf{W}_i \mathbf{x}_i^{\text{out}} + \mathbf{b}_i.$$

Since $W_\epsilon^\pi[i] \subseteq \mathbb{R}^{m_i \times n_i} \times \mathbb{R}^{n_i}$ is a product of intervals and therefore a connected set w.r.t. the Euclidean metric and since $h_i$ is continuous, the image of $W_\epsilon^\pi[i]$ under $h_i$ is also connected in $\mathbb{R}$. But note that

$$h_i(\mathbf{M}_i - \epsilon \cdot \mathbf{1}, \mathbf{m}_i - \epsilon \cdot \mathbf{1}) = (\mathbf{M}_i - \epsilon \cdot \mathbf{1})\mathbf{x}_i^{\text{out}} + (\mathbf{m}_i - \epsilon \cdot \mathbf{1})$$

---

[*]Equal Contribution.

35th Conference on Neural Information Processing Systems (NeurIPS 2021).

and
$$h_i(\mathbf{M}_i + \epsilon \cdot \mathbf{1}, \mathbf{m}_i + \epsilon \cdot \mathbf{1}) = (\mathbf{M}_i + \epsilon \cdot \mathbf{1})\mathbf{x}_i^{\text{out}} + (\mathbf{m}_i + \epsilon \cdot \mathbf{1}),$$

with $(\mathbf{M}_i - \epsilon \cdot \mathbf{1}, \mathbf{m}_i - \epsilon \cdot \mathbf{1}), (\mathbf{M}_i + \epsilon \cdot \mathbf{1}, \mathbf{m}_i + \epsilon \cdot \mathbf{1}) \in W_\epsilon^\pi[i]$. Thus, for the two points to be connected, the image set must also contain $\mathbf{x}_{i+1}^{\text{in}}$ which lies in between by eq. (1). Thus, there exists $(\mathbf{W}_i^*, \mathbf{b}_i^*) \in W_\epsilon^\pi[i]$ with $\mathbf{x}_{i+1}^{\text{in}} = \mathbf{W}_i^* \mathbf{x}_i^{\text{out}} + \mathbf{b}_i^*$, as desired.

For the input and the first hidden layer, from the BNN input layer constraint we know that

$$(\mathbf{M}_0 - \epsilon \cdot \mathbf{1})\mathbf{x}_{0,\text{pos}} + (\mathbf{M}_0 + \epsilon \cdot \mathbf{1})\mathbf{x}_{0,\text{neg}} + (\mathbf{m}_0 - \epsilon \cdot \mathbf{1}) \le \mathbf{x}_1^{\text{in}} \le (\mathbf{M}_0 + \epsilon \cdot \mathbf{1})\mathbf{x}_{0,\text{pos}} + (\mathbf{M}_0 - \epsilon \cdot \mathbf{1})\mathbf{x}_{0,\text{neg}} + (\mathbf{m}_0 + \epsilon \cdot \mathbf{1}).$$

Again, define $W_\epsilon^\pi[0]$ to be the set of all weight vectors between the input and the first hidden layer such that each weight value is distant from its mean by at most $\epsilon$. Consider a continuous function $h_0 : W_\epsilon^\pi[0] \to \mathbb{R}$ defined via
$$h_0(\mathbf{W}_0, \mathbf{b}_0) = \mathbf{W}_0 \mathbf{x}_0 + \mathbf{b}_0.$$
Let $\text{Msign}(\mathbf{x}_0)$ be a matrix of the same dimension as $\mathbf{M}_0$, with each column consisting of 1's if the corresponding component of $\mathbf{x}_0$ is nonnegative, and of $-1$'s if it is negative. Then note that

$$h_0(\mathbf{M}_0 - \epsilon \cdot \text{Msign}(\mathbf{x}_0), \mathbf{m}_0 - \epsilon \cdot \mathbf{1}) = (\mathbf{M}_0 - \epsilon \cdot \mathbf{1})\mathbf{x}_{0,\text{pos}} + (\mathbf{M}_0 + \epsilon \cdot \mathbf{1})\mathbf{x}_{0,\text{neg}} + (\mathbf{m}_0 - \epsilon \cdot \mathbf{1})$$

and

$$h_0(\mathbf{M}_0 + \epsilon \cdot \text{Msign}(\mathbf{x}_0), \mathbf{m}_0 + \epsilon \cdot \mathbf{1}) = (\mathbf{M}_0 + \epsilon \cdot \mathbf{1})\mathbf{x}_{0,\text{pos}} + (\mathbf{M}_0 - \epsilon \cdot \mathbf{1})\mathbf{x}_{0,\text{neg}} + (\mathbf{m}_0 + \epsilon \cdot \mathbf{1}).$$

Since $(\mathbf{M}_0 - \epsilon \cdot \text{Msign}(\mathbf{x}_0), \mathbf{m}_0 - \epsilon \cdot \mathbf{1}), (\mathbf{M}_0 + \epsilon \cdot \text{Msign}(\mathbf{x}_0), \mathbf{m}_0 + \epsilon \cdot \mathbf{1}) \in W_\epsilon^\pi[0]$, analogous image connectedness argument as the one above shows that there exist values $\mathbf{W}_0^*, \mathbf{b}_0^*$ of BNN weights such tha $(\mathbf{W}_0^*, \mathbf{b}_0^*) \in W_\epsilon^\pi[0]$, and such that $\mathbf{x}_1^{\text{in}} = \mathbf{W}_0^* \mathbf{x}_0 + \mathbf{b}_0^*$.

But now, collecting $\mathbf{W}_0^*, \mathbf{b}_0^*$ and $\mathbf{W}_i^*, \mathbf{b}_i^*$ for $1 \le i \le l - 1$ gives rise to a BNN weight vector $(\mathbf{W}^*, \mathbf{b}^*)$ which is contained in $W_\epsilon^\pi$. Furthermore, combining what we showed above with the constraints in $\Phi(\pi, \mathcal{X}_0, \mathcal{X}_u, \epsilon)$, we get that:

- $\mathbf{x}_0 \in \mathcal{X}_0$, $\mathbf{x}_l \in \mathcal{X}_u$, from the Input-output condition in $\Phi(\pi, \mathcal{X}_0, \mathcal{X}_u, \epsilon)$;
- $\mathbf{x}_i^{\text{out}} = \text{ReLU}(\mathbf{x}_i^{\text{in}})$ for each $1 \le i \le l - 1$, from the ReLU-encoding;
- $\mathbf{x}_1^{\text{in}} = \mathbf{W}_0^* \mathbf{x}_0 + \mathbf{b}_0^*$ and $\mathbf{x}_{i+1}^{\text{in}} = \mathbf{W}_i^* \mathbf{x}_i^{\text{out}} + \mathbf{b}_i^*$ for each $1 \le i \le l - 1$, as shown above.

Hence, $\mathbf{x}_l \in \mathcal{X}_u$ is the vector of neuron output values of $\pi_{\mathbf{W}^*, \mathbf{b}^*}$ on the input neuron values $\mathbf{x}_0 \in \mathcal{X}_0$, so as $(\mathbf{W}^*, \mathbf{b}^*) \in W_\epsilon^\pi$ we conclude that there exists a deterministic NN in $\{\pi_{\mathbf{w}, \mathbf{b}} \mid (\mathbf{w}, \mathbf{b}) \in W_\epsilon^\pi\}$ which is not safe. This concludes the proof. $\square$

**Theorem 2.** *If there exists a $W_\epsilon^\pi$-safe positive invariant, then each trajectory in $\textsf{Traj}_\epsilon$ is safe.*

*Proof.* Let $\textsf{Inv}$ be a $W_\epsilon^\pi$-safe positive invariant. Given a trajectory $(\mathbf{x}_t, \mathbf{u}_t)_{t=0}^\infty$ in $\textsf{Traj}_\epsilon$, we need to show that $\mathbf{x}_t \notin \mathcal{X}_u$ for each $t \in \mathbb{N}_{\ge 0}$. Since $\textsf{Inv} \cap \mathcal{X}_u = \emptyset$, it suffices to show that $\mathbf{x}_t \in \textsf{Inv}$ for each $t \in \mathbb{N}_{\ge 0}$. We prove this by induction on $t$.

The base case $\mathbf{x}_0 \in \textsf{Inv}$ follows since $\mathbf{x}_0 \in \mathcal{X}_0 \subseteq \textsf{Inv}$. As an inductive hypothesis, suppose now that $\mathbf{x}_t \in \textsf{Inv}$ for some $t \in \mathbb{N}_{\ge 0}$. We need to show that $\mathbf{x}_{t+1} \in \textsf{Inv}$.

Since the trajectory is in $\textsf{Traj}_\epsilon$, we know that the BNN weight vector $(\mathbf{w}_t, \mathbf{b}_t)$ sampled at the time-step $t$ belongs to $W_\epsilon^\pi$, i.e. $(\mathbf{w}_t, \mathbf{b}_t) \in W_\epsilon^\pi$. Thus, since $\mathbf{x}_t \in \textsf{Inv}$ by the induction hypothesis and since $\textsf{Inv}$ is closed under the system dynamics when the sampled weight vector is in $W_\epsilon^\pi$, it follows that $\mathbf{x}_{t+1} = f(\mathbf{x}_t, \mathbf{u}_t) = f(\mathbf{x}_t, \pi_{\mathbf{w}_t, \mathbf{b}_t}(\mathbf{x}_t)) \in \textsf{Inv}$. This concludes the proof by induction. $\square$

**Theorem 3.** *The loss function $\mathcal{L}$ is nonnegative for any neural network $g$, i.e. $\mathcal{L}(g) \ge 0$. Moreover, if $\textsf{Inv}$ is a $W_\epsilon^\pi$-safe positive invariant and $\mathcal{L}_{cls}$ the 0/1-loss, then $\mathcal{L}(g^{\textsf{Inv}}) = 0$. Hence, neural networks $g^{\textsf{Inv}}$ for which $\textsf{Inv}$ is a $W_\epsilon^\pi$-safe positive invariant are global minimizers of the loss function $\mathcal{L}$ when $\mathcal{L}_{cls}$ is the 0/1-loss.*

*Proof.* Recall, the loss function $\mathcal{L}$ for a neural network $g$ is defined via

$$\mathcal{L}(g) = \frac{1}{|D_{\text{spec}}|} \sum_{(\mathbf{x}, y) \in D_{\text{spec}}} \mathcal{L}_{\text{cls}}\big(g(\mathbf{x}), y\big) + \lambda \frac{1}{|D_{\text{ce}}|} \sum_{(\mathbf{x}, \mathbf{x}') \in D_{\text{ce}}} \mathcal{L}_{\text{ce}}\big(g(\mathbf{x}), g(\mathbf{x}')\big), \tag{2}$$

where $\lambda$ is a tuning parameter and $\mathcal{L}_{\text{cls}}$ a binary classification loss function, e.g. the 0/1-loss $\mathcal{L}_{0/1}(z, y) = \mathbb{1}[\mathbb{1}[z \geq 0] \neq y]$ or the logistic loss $\mathcal{L}_{\log}(z, y) = z - z \cdot y + \log(1 + \exp(-z))$ as its differentiable alternative. The term $\mathcal{L}_{\text{ce}}$ is the counterexample loss which we define via

$$\mathcal{L}_{\text{ce}}(z, z') = \mathbb{1}[z > 0]\,\mathbb{1}[z' < 0]\,\mathcal{L}_{\text{cls}}(z, 0)\,\mathcal{L}_{\text{cls}}(z', 1). \tag{3}$$

The fact that $\mathcal{L}(g) \geq 0$ for each neural network $g$ follows immediately from the fact that summands in the first sum in eq. (2) are loss functions which are nonnegative, and summands in the second sum are products of indicator and nonnegative loss functions and therefore also nonnegative.

We now show that, if $\mathcal{L}_{\text{cls}}$ is the 0/1-loss, $\mathcal{L}(g^{\text{Inv}}) = 0$ whenever Inv is a $W_\epsilon^\pi$-safe positive invariant, which implies the global minimization claim in the theorem. This follows from the following two items:

- For each $(\mathbf{x}, y) \in D_{\text{spec}}$, we have $\mathcal{L}_{\text{cls}}(g^{\text{Inv}}(\mathbf{x}), y) = 0$. Indeed, for $(\mathbf{x}, y)$ to be added to $D_{\text{spec}}$ in Algorithm 1, we must have that either $\mathbf{x} \in \mathcal{X}_0$ and $y = 1$, or that $\mathbf{x} \in \mathcal{X}_u$ and $y = 0$. Thus, since Inv is assumed to be a $W_\epsilon^\pi$-safe positive invariant, $g^{\text{Inv}}$ correctly classifies $(\mathbf{x}, y)$ and the corresponding loss is 0.
- For each $(\mathbf{x}, \mathbf{x}') \in D_{\text{ce}}$ we have $\mathcal{L}_{\text{ce}}(g^{\text{Inv}}(\mathbf{x}), g^{\text{Inv}}(\mathbf{x}')) = 0$. Indeed, since

$$\mathcal{L}_{\text{ce}}(g^{\text{Inv}}(\mathbf{x}), g^{\text{Inv}}(\mathbf{x}')) = \mathbb{1}[g^{\text{Inv}}(\mathbf{x}) > 0]\,\mathbb{1}[g^{\text{Inv}}(\mathbf{x}') < 0]\,\mathcal{L}_{\text{cls}}(g^{\text{Inv}}(\mathbf{x}), 0)\,\mathcal{L}_{\text{cls}}(g^{\text{Inv}}(\mathbf{x}'), 1),$$

for the loss to be non-zero we must have that $g^{\text{Inv}}(\mathbf{x}) \geq 0$ and $g^{\text{Inv}}(\mathbf{x}) < 0$. But this is impossible since Inv is assumed to be a $W_\epsilon^\pi$-safe positive invariant and $(\mathbf{x}, \mathbf{x}')$ was added by Algorithm 1 as a counterexample to $D_{\text{ce}}$, meaning that $\mathbf{x}'$ can be reached from $\mathbf{x}$ by following the dynamics function and sampling a BNN weight vector in $W_\epsilon^\pi$. Therefore, by the closedness property of $W_\epsilon^\pi$-safe positive invariants when the sampled weight vector is in $W_\epsilon^\pi$, we cannot have both $g^{\text{Inv}}(\mathbf{x}) \geq 0$ and $g^{\text{Inv}}(\mathbf{x}) < 0$. Hence, the loss must be 0.

$\square$

**Theorem 4.** *If the verifier in Algorithm 1 shows that constraints in three checks are unsatisfiable, then the computed Inv is indeed a $W_\epsilon^\pi$-safe positive invariant. Hence, Algorithm 1 is correct.*

*Proof.* The fact that the first check in Algorithm 1 correctly checks whether there exist $\mathbf{x}, \mathbf{x}' \in \mathcal{X}_0$ and a weight vector $(\mathbf{w}, \mathbf{b}) \in W_\epsilon^\pi$ such that $\mathbf{x}' = f(\mathbf{x}, \pi_{\mathbf{w}, \mathbf{b}}(\mathbf{x}))$ with $g^{\text{Inv}}(\mathbf{x}) \geq 0$ and $g^{\text{Inv}}(\mathbf{x}') < 0$ follows by the correctness of our encoding in Section 4.1, which was proved in Theorem 1. The fact that checks 2 and 3 correctly check whether for all $\mathbf{x} \in \mathcal{X}_0$ we have $g^{\text{Inv}}(\mathbf{x}) \geq 0$ and for all $\mathbf{x} \in \mathcal{X}_u$ we have $g^{\text{Inv}}(\mathbf{x}) < 0$, respectively, follows immediately from the conditions they encode.

Therefore, the three checks together verify that (1) Inv is closed under the system dynamics whenever the sampled weight vector is in $W_\epsilon^\pi$, (2) Inv contains all initial states, and (3) Inv contains no unsafe states. As these are the 3 defining properties of $W_\epsilon^\pi$-safe positive invariants, Algorithm 1 is correct and the theorem claim follows. $\square$

# B  Safe exploration reinforcement learning algorithm

Algorithm 1 shows our sketch of how standard RL algorithms, such as policy gradient methods and deep Q-learning, can be adapted to a safe exploration setup by using the safe weight sets computed by our method.

# C  Experimental details

In this section, we describe the details of our experimental evaluation setup. The code and pre-trained network parameters are attached in the supplementary materials. Each policy is a ReLU network consisting of three layers. The first layer represents the input variables, the second one is a hidden layer with 16 neurons, and the last layer are the output variables. The size of the first and the last layer is task dependent and is shown in Table 1. The $W_\epsilon^\pi$-safe positive invariant candidate network differs from the policy network in that its weights are deterministic, it has a different number of hidden units and a single output dimension. Particularly, the invariant networks for the linear dynamical system

**Algorithm 1** Safe Exploration Reinforcement Learning

---

**Input** Initial policy $\pi_0$, learning rate $\alpha$, number of iterations $N$
**for** $i \in 1, \ldots N$ **do**
    $\epsilon \leftarrow$ find safe $\epsilon$ for $\pi_{i-1}$
    collect rollouts of $\pi_{i-1}$ with rejection sampling $\epsilon$
    compute $\nabla \mu_{i-1}$ for parameters $\mu_{i-1}$ of $\pi_{i-1}$ using DQN/policy gradient
    $\mu_i \leftarrow \mu_{i-1} - \alpha \nabla \mu_{i-1}$ (gradient descent)
    project $\mu_i$ back to interval $[\mu_{i-1} - \epsilon, \mu_{i-1} + \epsilon]$
**end for**
**Return** $\pi_N$

---

| Experiment | Input dimension | Hidden size | Output dimension |
|---|---|---|---|
| Linear dynamical system | 2 | 16 | 1 |
| Inverted pendulum | 2 | 16 | 1 |
| Collision avoidance | 3 | 16 | 3 |

Table 1: Number of dimensions of the policy network for the three experiments.

and the inverted pendulum have 12 hidden units, whereas the invariant network for the collision avoidance task has 32 neurons in its hidden layer. The policy networks are trained with a $\mathcal{N}(0, 0.1)$ (from second layer on) and $\mathcal{N}(0, 0.05)$ (all weights) prior for the Bayesian weights, respectively. MILP solving was performed by Gurobi 9.03 on a 4 vCPU with 32GB virtual machine.

**Linear dynamical system** The state of the linear dynamical system consists of two variables $(x, y)$. The update function takes the current state $(x_t, y_t)$ with the current action $u_t$ and outputs the next states $(x_{t+1}, y_{t+1})$ governed by the equations

$$y_{t+1} = y_t + 0.2 \cdot \mathrm{clip}_{\pm 1}(u_t)$$
$$x_{t+1} = x_t + 0.3 y_{t+1} + 0.05 \cdot \mathrm{clip}_{\pm 1}(u_t),$$

where the function $\mathrm{clip}_{\pm 1}$ is defined by

$$\mathrm{clip}_{\pm z}(x) = \begin{cases} -z & \text{if } x \leq -z \\ z & \text{if } x \geq z \\ x & \text{otherwise.} \end{cases}$$

The set of unsafe states is defined as $\{(x, y) \in \mathbb{R}^2 \mid |(x, y)|_\infty \geq 1.2\}$, and the initial states as $\{(x, y) \in \mathbb{R}^2 \mid |(x, y)|_\infty \leq 0.6\}$.

**Inverted pendulum** The state of the inverted pendulum consists of two variables $(\theta, \dot{\theta})$. The non-linear state transition is defined by

$$\dot{\theta}_{t+1} = \mathrm{clip}_{\pm 8}\left(\dot{\theta}_t + \frac{-3g \cdot \mathrm{angular}(\theta_t + \pi)}{2l} + \delta_t \frac{7.5 \mathrm{clip}_{\pm 1}(u_t)}{(m \cdot l^2)}\right)$$
$$\theta_{t+1} = \theta_t + \dot{\theta}_{t+1} * \delta_t,$$

where $g = 9.81, l = 1, \delta_t = 0.05$ and $m = 0.8$ are constants. The function angular is defined using the piece-wise linear composition

$$\mathrm{angular}(x) = \begin{cases} \mathrm{angular}(x + 2\pi) & \text{if } x \leq \pi/2 \\ \mathrm{angular}(x - 2\pi) & \text{if } x > 5\pi/2 \\ \frac{x - 2\pi}{\pi/2} & \text{if } 3\pi/2 < x \leq 5\pi/2 \\ 2 - \frac{x}{\pi/2} & \text{if } \pi/2 < x \leq 3\pi/2. \end{cases}$$

The set of initial states are defined by $\{(\theta, \dot{\theta}) \in \mathbb{R}^2 \mid |\theta| \leq \pi/6 \text{ and } |\dot{\theta}| \leq 0.2\}$. The set of unsafe states are defined by $\{(\theta, \dot{\theta}) \in \mathbb{R}^2 \mid |\theta| \geq 0.9 \text{ or } |\dot{\theta}| \geq 2\}$.

**Collision avoidance**    The state of the collision avoidance environment consists of three variables $(p_x, a_x, a_y)$, representing the agent's vertical position and the vertical and the horizontal position of an intruder. The intruder moves toward the agent, while the agent's vertical position must be controlled to avoid colliding with the intruder. The particular state transition is given by

$$
\begin{aligned}
p_{x,t+1} &= p_{x,t} + u_t \\
a_{x,t+1} &= a_{x,t} \\
a_{y,t+1} &= a_{y,t} - 1.
\end{aligned}
$$

Admissible actions are defined by $u_t \in \{-1, 0, 1\}$. The set of initial states are defined as $\{(p_x, a_x, a_y) \in \mathbb{Z}^3 \mid |p_x| \leq 2 \text{ and } |a_x| \leq 2 \text{ and } a_y = 5\}$. Likewise, the set of unsafe states are given by $\{(p_x, a_x, a_y) \in \mathbb{Z}^3 \mid |p_x - a_x| \leq 1 \text{ and } a_y = 5\}$.