# OpenReview forum: "Infinite Time Horizon Safety of Bayesian Neural Networks"
_NeurIPS.cc/2021/Conference — NeurIPS 2021 Poster_

### Official Review · Reviewer_dPLB · 2021-07-02

**Rating:** 5
**Confidence:** 4

**Summary:**

The paper considers the problem of safely controlling a known, discrete-time system with a stochastic BNN policy with ReLU activations where randomness enters through the policy's weights. Safety is defined in terms of avoiding a set of states that are known to be safe over an infinite time horizon. The paper first considers the problem of determining whether BNN outputs lie in a set of unsafe actions. To this end, the paper proposes to approximate weights by bounding boxes in a set W_epsilon and verifies safety through a corresponding mixed-interger linear program. While the BNN's distribution typically has unbounded support, for any verified epsilon it can be made safe through rejections sampling.

The developed methodology is then used for the safe control problem, where a neural network is used together with a classification problem in order to learn an invariant set under the policy. The algorithm proceeds iteratively by fitting the neural network and then finding counter-examples through a mixed-integer program that are then added to the training set. The result is a trained classifier for safe/unsafe states for a given policy and value epsilon.

The approach is evaluated on two linear and one piecewise-linear dynamic systems with action dimension one and state dimensions between one and three, showing that for these problems the proposed method can verify safety within a reasonable runtime.

**Ethical Concerns:**

They are addressed at the end of the conclusion

**Limitations And Societal Impact:**

Limitations
-----------
Societal impact is addressed, but limitations are not discussed sufficiently as mentioned in the main review
- Ignoring potential correlations between BNN weights
- Box-approximations to non-Gaussian distributions in general
- known, deterministic dynamics. Piecewise-linearity
- Scalability of the MIPs

**Main Review:**

The paper is clearly written and easy to follow. There are some minor inconsistencies in variable naming that I list at the end of the review. The problem setting that the paper sets out to solve is interesting and safety is certainly a relevant concern for neural networks. However, it is not clear how one would obtain a policy that has uncertain weights, given that it is usually the environment that is unknown. The paper could better motivate why this approach is significance of the problem.

The methodology in the paper focuses on two problems. The first one considers verifying that the output of a network does not lie withing a given set, which is verified using a MILP. While I'm not sure the particular problem of encoding weight uncertainty in the MILP has been considered, since epsilon is assumed to be known the structure is very close to previous constructions. As such I see the main contribution of the paper in how the second problem is solved.

One of my biggest concerns is that the paper over-promises initially and is not explicit about all the limitations. In the abstract the paper promises to provide infinite-horizon safety for general BNN policies in reinforcement learning. This scope the reduces over the course of the paper:
- First environments are restricted to be deterministic and fully known, which means that we are no longer in the reinforcement learning setting, but instead dealing with control (known environment).
- Next the BNN policy is replaced by a policy where each weight is allowed to vary independently within a Box. This is reasonable for a prior BNN (i.e., all weights are sampled iid), but any kind of trained/posterior weights are generally strongly correlated. For this setting assuming independent bounding boxes can potentially be a strong over-approximation. At the same time there is a statement that we can verify the full BNN support by letting epsilon -> infinity, which is essentially allowing weights to shift between +/-infinity, arguably not a useful scenario.
- Lastly, to find counter-examples the paper proposes a complex non-convex problem, which as far as I can tell can only be solved using the introduced methodology if the environment dynamics are piecewise-linear. This also shows in the experiments, where the piecewise-linear inverted pendulum is essentially linearized at two points, one of which seems to excluded through the constraint set.

Together these are strong limitations imposed on the approach that would need to be discussed and evaluated in the paper and should be clear from the beginning.

The experiments are restricted to synthetic (policy is pre-defined and weight uncertainty is set to low, manually defined values), simple, and low dimensional problems (state dimension < 3, action dimension = 1, and 2-layer networks with 16 units). There is no discussion on the scalability of the approach and there are no comparisons the existing method (e.g., the sampling-based methods mentioned in the related work). Additionally, only single-runs are reported, even though the training process for the neural network classifier is likely to be noisy so that error bars on compute time and performance would be appropriate. In my opinion indicated this as [N/A] in the checklist is not appropriate here.

### Summary

Overall between the lack of motivation for the setting (where does my BNN policy come from), the limited discussion of the requirements, limitations and assumptions, together with the limited experiments without comparison against other methods make this paper, in my opinion, not ready for NeurIPS in its current form.

### Stylistic and minor comments

- Fig. 1 should not be wrapped in the text, since NeurIPS has a single-column format.
- X_u is defined as both the set of unsafe actions and states depending on context. Maybe differentiating between these two sets notationally could improve clarity.
- pi is defined as a composition of ReLU layers, but presumably there's no ReLU on the output
- There is some confusion between the number of layers in Sec. 4.1. In the text it is "k", while in the system constraints it is "l"
- In the system constraints, x_l is not defined in terms of x_{l-1}/x_{k-1}.
- The paper checklist 4b) indicates that licences of cited assets are provided (e.g., gurobi) are stated, but I could not find any mention of them in the text or appendix.

### Post-rebuttal

Increased score after reading the author response, but the concerns about restrictive assumptions, scalability, and practical relevance remain.

**Time Spent Reviewing:**

3

---

> ### Author Response · Authors · 2021-08-06
> **We thank the reviewer for the valuable feedback and comments**
>
> We thank the reviewer for the valuable feedback and comments.
>
> The reviewer mentions a set of limitations of our approach. We clarify below that some of the limitations mentioned are incorrect and we will revise our paper to avoid the misinterpretation. The detailed response is below:
>
> **Deterministic dynamics**: Many environments of the OpenAI gym have deterministic dynamics with randomness coming only from the initial state. Using a BNN policy in such deterministic environments allows us to safely explore the environment, as well as reason about the uncertainty of the optimal action and has therefore valid practical use cases. Moreover, specifically to our work, modelling the environment by deterministic dynamics is a common assumption made in RL safety and stability research [1,2,3]. We agree with the reviewer that stochasticity in the environment’s dynamics poses an interesting direction of future work.
>
> **Correlated weights** Our approach is also applicable to BNNs with weights having correlated multivariate distributions. However, sampling each component of the random weight vector independently is a standard practice in BNN literature [4], due to practical reasons, e.g. a full covariance matrix of a ResNet20 would have trillions of entries. Note that, although the weights are sampled independently, the parameters of their distributions (and, in particular, their posteriors’ means) are obviously dependent after training. As our method puts the box-support around each weight’s posterior’s mean, our method captures this dependency.
>
> **Box-support sets** Unimodal distributions (Gaussian, Laplace, and Cauchy) are the most commonly used priors for BNN weights [4,5]. For such distributions our box-supports around the mean provide a plausible size-vs-probability mass ratio. Moreover, in Section 6 we state our choice as a limitation for BNNs with multi-modal priors, and relaxing this restriction is an important direction of future work. We will highlight this in the paper more clearly.
>
> **Infinite epsilon is not useful** The reviewer argues that having +- infinite as an epsilon is not a useful scenario. This is exactly what we argue in Figure 1 and is the underlying motivation of our approach.
> Our comment at the end of Section 3 is for completeness, to emphasize that we could *in theory* use our approach to verify safety of BNN policies with epsilon=infty.
>
> **Comparison to sampling-based approaches** As mentioned in the paper, sampling-based approaches can only reason about finite time horizons and are therefore inapplicable to certifying infinite time horizon safety.
>
> **Missing error bars** The reviewer criticizes missing error bars and issues regarding the reproducibility of our experiments. We want to restate that we will publicly release the entire source code of our experiments along with the paper.
>
> **Only piecewise linear systems** Regarding the restriction to piecewise nonlinear environments: As outlined in Section 4.1, our theoretical framework holds for constraint solving via non-linear real arithmetic SMT-solving (supporting other types of nonlinearities) as well. We implemented our experiments with MILP solving for practical reasons, namely well established neural network encodings and decent scalability reported in the literature.
>
> **Scalability** Any verification method for neural networks (even more so for neural networks in feedback loops) suffers from scalability limitations due to the underlying complexity class [6]. As pointed out by Review Md7z this applies to the whole research area and not in particular to our approach. We mention in checklist 1b) that the constraint solving step poses a scalability bottleneck on our approach. We will highlight this more clearly in the conclusion (Section 6) of the final version of the paper. In Section 4.1, we also included a discussion on the trade-off between scalability and generality of different constraint solving techniques.
>
> [1] Berkenkamp et al. Safe Model-based Reinforcement Learning with Stability Guarantees. NeurIPS 2017
> [2] Richards et al. The Lyapunov Neural Network: Adaptive Stability Certification for Safe Learning of Dynamical Systems. CoRL 2018
> [3] Chang et al. Neural Lyapunov Control. NeurIPS 2019
> [4] Wenzel et al. How Good is the Bayes Posterior in Deep Neural Networks Really? ICML 2020
> [5] Blundell et al. Weight Uncertainty in Neural Network. ICML 2015
> [6] Katz et al. Reluplex: An Efficient SMT Solver for Verifying Deep Neural Networks. CAV 2017
>
>
> We thank the reviewer for pointing out the stylistic and minor comments and will update the paper accordingly.

---

### Official Review · Reviewer_Md7z · 2021-07-13

**Rating:** 6
**Confidence:** 3

**Summary:**

We extend verifications of (piece-wise linear or SMT compliant) neural networks to bayesian neural networks.

**Limitations And Societal Impact:**

this work addresses them adequately

**Main Review:**

### Clarity
This paper is clear, I have no problem following it

### Originality
I think this work is predictable, as just adding another quantifier "\forall range of values in BNN weights" is SMT compliant, and most verification techniques to verify deterministic NN can be ported over more or less directly. However this does not diminish the contribution of this work: Being able to verify a non-deterministic agent (such as one used in exploration and learning of an environment) behaviour in an infinite time-horizon system is a very important problem, especially in the domain of safe exploration that this paper considers.

### Quality
The proofs on using invariants to guarantee safety in the infinite horizon setting is a good and sound. The key contribution to me is how this paper bypass the undecidable hardness of generating an invariant by using learning to _propose_ an invariant, then using a decision procedure to _verify_ it, making it tractable. The decision (the same as other NN+SMTVerification works) is made to treat distributions as ranges, then applying rejection sampling makes this approach compliant to off-the-shelf SMT/Linear solvers, which has an easier time encoding ranges of values and existence of values as opposed to reasoning over distributions.

### Significance
The complaint here is the typical one for verification: How does this scale to a realistic scenario? Presumably given a sufficiently difficult environment, with non-linear dynamics or a sufficiently complex neural network (say a transformer with some nonlinearity), this approach will be not applicable. In a sense, if the best example is a 2D 2 dot AND 1 dot is restricted to a line (not even on an arbitrary spline, presumably spline breaks SMT?) simulation, how do we proceed in taking this technique and apply it to a more complex domain (such as 2D 5 dots, maybe 1 is a car and other 4 pedestrians, with some constraints on the paths they can take)? While I am not concerned that the results present in this work are preliminary and stilted, what are some avenues forward that you can see (assuming solver techs are not going to improve)?

Similarly, "However, all of the constraint-solving techniques we discuss were already used in previous work for RL safety and stability verification" is bit of a kicking the can down the road for me.

going to keep my original score after reading the review, as the same message stands: this is a good paper, but the best example is 2D with 2 dots and 1 dot is restricted to a line.

**Time Spent Reviewing:**

1 hour

---

> ### Author Response · Authors · 2021-08-06
> **We thank the reviewer for the valuable feedback and comments**
>
> We thank the reviewer for the valuable feedback and comments.
>
> We fully agree with the reviewer that scalability is the central challenge of neural network verification. However, as mentioned by the reviewer, this criticism applies to the research field in general and not just our paper specifically.
>
> Our work provides, for the first time, a theoretical framework for reasoning about safety of Baysian neural networks over the infinite time horizon. A promising research direction on improving the scalability of our approach would be to speed up the limiting constraint solving step by a faster gradient based optimization [1].
> A second direction would be to incorporate the constraint solving step already in the training procedure, which has been done in [2] for verifying robustness of feed-forward NNs.
> These are directions for future works as adapting [1] and [2] to feedback systems are non-trivial tasks. However, these two works hint that improving the scalability beyond MILP/SMT based techniques is possible.
>
> [1] Henriksen and Lomuscio. Efficient Neural Network Verification via Adaptive Refinement and Adversarial Search. ECAI 2020
> [2] Zhang et al. Towards Stable and Efficient Training of Verifiably Robust Neural Networks. ICLR 2020

---

### Official Review · Reviewer_RfGA · 2021-07-16

**Rating:** 7
**Confidence:** 3

**Summary:**

In this paper the authors consider the problem of verifying safety in BNNs.  At a high level, they seek a way to compute a set of safe weight sets of a BNN such that its outputs can not fall within some “unsafe set.” They show doing so for two contexts, the first aiding the second. In the first context they take a standard feed forward BNN, while in the second they consider a closed loop system under some dynamics where the policy at each time point is implemented by a BNN.

**Limitations And Societal Impact:**

The authors adequately addressed the limitations and potential negative societal impact of their work.

**Main Review:**

# Strengths
The authors do a good job of why this is an important problem in the literature; i.e. safety critical applications and in the context of exploration RL where a policy would like to be designed such that we don’t wind up in “unsafe” states.  Literature review is also highly appreciated and does a good job of further motivating the problem and the authors contributions. By starting with the problem of vanilla feed forward BNNs the extension to closed loop systems governed by some dynamics is made more natural.

Outlining several ways of solving the constraints (MILP, Reluplex, NRA-SMT) demonstrates that there are several ways of approaching the problem each with their own strengths and weaknesses; and also that there exist methods that make it viable to do so.

I find the learner and verifier algorithm for learning safe positive invariants creative as well.  Though complicated (which I do not find a downside) as far as I can tell the logic is sound and trying to give the correct intuition was also appreciated.

Choice of experiments seem sufficient as well; the paper is fairly theory heavy so having some simple experiments to show the efficacy of the proposed approach was good. Also glad to see runtimes listed as this would have been one of my bigger questions.



# Other/Questions
I am wondering how is epsilon usually chosen? I’m not sure this is an appropriate question but perhaps a little more intuition regarding this could have been given in the paper. As a follow up to this I would also like to ask why a uniform epsilon across all weights/biases (I see this was addressed in the conclusion but Im still curious)?

As this does not seem like a well explored problem in the literature I think the work presented here is a good first step; as such systems become more prominent guaranteeing safety critical aspects will become increasingly more necessary.

# POST REBUTTAL
Having read through the reviews as well as the authors responses I am satisfied and leave my score unchanged. To the authors, thanks for some clarifications about eps. I also wanted to say that I think the suggestions offered by axJp seem like they will be a great addition to the final paper.

**Time Spent Reviewing:**

7

---

> ### Author Response · Authors · 2021-08-06
> **We thank the reviewer for the valuable feedback and comments**
>
> We thank the reviewer for the valuable feedback and comments.
>
> Concerning the questions raised by the reviewer:
>
> **Choice of $\epsilon$.** In our experiments, we first certified safety for a small value of $\epsilon$ and then iteratively increased $\epsilon$ until we could either no longer certify safety or if the timeout was reached. Note that, in each iteration, our Algorithm 1 does not start from scratch but is initialized with the $g_{inv}$ and $D_{spec}$ from the previous successful iteration. We will clarify this in the final version of the paper.
>
> **Uniform value of $\epsilon$.** Our theoretical results and BNN encoding are also applicable to the case when epsilons are non-uniform over weights and biases.
> However, we don’t have a principled way to start with some value of epsilons and then iteratively increase them in a non-uniform way. In a hypothetical example, if we had two different values eps1 and eps2 for different weights, increasing eps1 and eps2 at the same time could be unsafe whereas increasing only one of the two is safe. So it is not immediately clear which one should be increased (because the set of all products of intervals does not define a total order under set inclusion). This is why in our experimental setup we use uniform values of epsilons over weights and biases (epsilons of the form $\epsilon$ * variance(weight_i) for weight_i, to be precise).
> Investigating a way to compute non-uniform safe weight sets is an interesting direction of future work and related to the general future direction of relaxing the current form of safe weight sets which we mention in Section 6.

---

### Official Review · Reviewer_axJp · 2021-07-17

**Rating:** 7
**Confidence:** 3

**Summary:**

This paper considers a safety verification problem for a BNN policy over an infinite time horizon, in the setting of a discrete-time closed-loop dynamical system. Since most BNN policies do not satisfy safety properties due to the unbounded supports of the BNNs’ weights, the paper considers the problem of finding an appropriate subset (called safe weight sets) of such an unbounded support which makes the desired safety properties hold. This problem can be viewed as a generalization of the original safety verification problem. The paper solves the generalized problem by computing a safe inductive invariant, which is represented by a separate NN, searched by a training procedure, and verified by a separate verifier module. The proposed method is evaluated on several benchmarks, and produces promising results.


**Limitations And Societal Impact:**

For limitations, refer to the main review. Societal impact is discussed in the paper.

**Main Review:**

## Strengths.

(1) This paper considers an important, interesting problem. In particular, it introduces a new problem of computing safe weight sets, which generalizes the original safety verification problem, to make the original verification problem produce more meaningful results. Further, a solution to the new problem can allow us to make a BNN policy safe via rejection sampling, which can be used importantly in safe-critical settings.

(2) The following parts of the proposed method look novel to me: using a NN to represent a safe inductive invariant, and casting the search problem for a safe inductive invariant into a NN learning problem. Note that most other parts of the method (considering inductive invariants, applying a verifier, and utilizing counterexamples from the verifier) are somewhat standard at least in the programming languages area.

(3) The proposed method for the new generalized problem is shown to perform well on several benchmarks.

(4) The paper is overall well-written (except for two specific points described below). Also the technical contents of the paper are sound.

## Weaknesses.

(1) The writing needs a bit more improvement. First, throughout the paper I got the impression that safe weight sets are automatically searched or learned by the proposed method, but this turns out not to be the case---the proposed method assumes a particular safe weight set (namely the product of intervals of radius \epsilon), and does not give any efficient way of finding safe weight sets. This point should be clarified in the paper.

Second, according to the experimental results, the bootstrap trick seems to be critical to good performance but it is not explained in detail in the text. Some more details on the bootstrap trick should be provided.

(2) The paper has two main limitations. First, as mentioned in Sec 6, the paper considers only a simple form of safe weight sets. Relaxing this restriction would be an important future work. Second, finding or learning safe weight sets automatically and efficiently is another important problem not resolved by this paper---in the experiments different \epsilon values are simply tried. Even with the two limitations, I think this paper still maintains its significance as it provides first steps towards more practical safety verification of BNNs.

## Questions & Comments.

- Can you show that the verified \epsilon values in Table 1 (e.g., 2\sigma) is (or close to) maximal? That is, can you show concretely that a large \epsilon value indeed makes \pi unsafe? If possible, that would strengthen the experimental results.

- Why do the experiments use two different BNN policies? Was there any particular criteria to choose the two particular BNNs used in the paper?

- Line 131-132. Several “U”s here should be in \mathcal.
- Line 168. Provide the type of \pi (i.e., \pi : X \to X), as this \pi is different from \pi : X \to D(U) in the above.
- From Line 201. Several “l”s in the subscript of x (e.g., x_l) should be “k”.
- Line 210. “x^out_0” should be “x_0,pos”.
- Line 219, 221. “Equations 3-4” and “equation 5” do not exist in the above equations.

-----

**Updates after author response.** Thank the authors for answering my questions. I am still positive on the paper and will keep the same score.

**Time Spent Reviewing:**

8

---

> ### Author Response · Authors · 2021-08-06
> **We thank the reviewer for the valuable feedback and comments**
>
> We thank the reviewer for the valuable feedback and comments. We start by providing clarifications that were pointed out by the reviewer, and then proceed to the questions that were raised by the reviewer.
>
> **Computation of safe weight sets and the choice of $\epsilon$.** Problems 1 and 2 assume a given value of $\epsilon$ for which $\pi$’s safety needs to be verified. In the experiments, in order to compute such a value of $\epsilon$ and therefore a safe weight set, we started by certifying safety for a small $\epsilon$ and then iteratively increased it until we reached a value that cannot be certified or until the timeout was reached. We note that, in each iteration, our Algorithm 1 does not start from scratch but is initialized with the $g_{inv}$ and $D_{spec}$ from the previous successful iteration. We will clarify this in the final version of the paper.
>
> **Bootstrap trick.** In the bootstrapping mode, we initialize the positive samples of $D_{spec}$ not just with $X_0$ but also with states reachable from $X_0$, i.e. with rollouts of the policy. Our theory applies to both modes (with and without bootstrapping), however as the reviewer mentioned, in practice the bootstrapping yields significant improvements. We will mention this in the experimental section more clearly.
>
> **Refuting large $\epsilon$.** This is a very interesting question. A trivial way to certify that $\pi$ is unsafe for a given $\epsilon$ is to sample rollouts until we find a rollout that reaches the unsafe set. However, such an approach is obviously only suitable for refuting safety over a finite time horizon. One of the key features of our work is that we consider infinite time horizon safety. We are currently not aware of a plausible approach for refuting infinite time horizon safety, and think that it is an exciting direction of future work. We will point out falsifying infinite time horizon safety as a future direction.
> To compute the maximal $\epsilon$ for which $\pi$ can be verified to be safe via our method, one can modify our iterative procedure in point 1 with a binary or an exponential search on $\epsilon$.
>
> **Two BNN policies in the experiments.** In our BNN encoding in Section 4.1, we show that encoding of the BNN input layer requires additional constraints and extra care, since we do not know the signs of input neuron values. Hence, in our experiments we train (1) a BNN whose all neuron weights are Bayesian and (2) a BNN whose all hidden layer neuron weights are Bayesian, in order to study how the encoding of the input layer affects the safe set computation. While we explain how each BNN policy is trained in the experimental setup, we agree that the motivation for this is missing, so we will include a remark that the motivation for training two such BNNs is coming from our BNN encoding in Section 4.1.
>
> **Other comments.** Thank you for the useful comments, we will address these in the final version.

---

### Decision · Program_Chairs · 2021-09-27

**Decision:**

Accept (Poster)

**Comment:**

The manuscript provides a theoretical framework and analysis for verifying safety in BNNs. Reviewers generally agree that it is well written and novel. Reviewers have also noted limitations and strong assumptions, and I believe the rebuttal process has been instrumental to improving the manuscript. I strongly suggest making the proposed changes in the final manuscript.